# Communication Systems Performance at mm and THz as a Function of a Rain Rate Probability Density Function Model

**DOI:** 10.3390/s22166269

**Published:** 2022-08-20

**Authors:** Judy Kupferman, Shlomi Arnon

**Affiliations:** 1Department of Electrical and Computer Engineering, Ben-Gurion University of the Negev, Beer Sheva 8441405, Israel; 2Center for Quantum Science and Technology, Ben-Gurion University of the Negev, Beer Sheva 8441405, Israel

**Keywords:** atmospheric propagation, communication system performance, attenuation, communication

## Abstract

6G is already being planned and will employ much higher frequencies, leading to a revolutionary era in communication between people as well as things. It is well known that weather, especially rain, can cause increased attenuation of signal transmission for higher frequencies. The standard methods for evaluating the effect of rain on symbol error rate are based on long-term averaging. These methods are inaccurate, which results in an inefficient system design. This is critical regarding bandwidth scarcity and energy consumption and requires a more significant margin of effort to cope with the imprecision. Recently, we have developed a new and more precise method for calculating communication system performance in case of rain, using the probability density function of rain rate. For high rain rate (above 10 mm/h), for a typical set of parameters, our method shows the symbol error rate in this range to be higher by orders of magnitude than that found by ITU standard methods. Our model also indicates that sensing and measuring the rain rate probability is important in order to provide the required bit error rate to the users. This will enable the design of more efficient systems, enabling design of an adaptive system that will adjust itself to rain conditions in such a way that performance will be improved. To the best knowledge of the authors, this novel analysis is unique. It can constitute a more efficient performance metric for the new era of 6G communication and prevent disruption due to incorrect system design.

## 1. Introduction

In the next decade, 6G communication will come into use. Nevertheless, it is already in the planning stage and is expected to revolutionize not only communication systems but all aspects of life and society, from innovative personal technology to networks in previously inaccessible areas. While the exact characteristics are not yet finalized, they will undoubtedly include speed and low latency, with a peak data rate on the order of 10 terabits per second, universal connection with low delay, and high reliability. Frequencies will reach the terahertz range, and the FCC has already taken preliminary steps toward a terahertz wave spectrum [1,2,3,4]. Eventually, 6G will range over the world in all kinds of areas and climates, and the high data rates and need for universal coverage will require supreme accuracy [5,6]. Primary applications include autonomous vehicle communication, the Internet of Things (IoT), medical instruments, and areas in which precision is of overreaching importance. Thus, 6G will have an imperative need for accurate calculation of the probability for error in symbol transmission.

The 6G system will find use both on satellite-ground links and on the ground. As higher frequencies are used, weather effects increasingly affect the signal [3,7,8]. Rain is the leading cause of errors regarding communication system performance concerning weather effects at a high frequency of mm and THz. Given the average rain rate R, the attenuation can be found as a function of R. Estimation of rain rate using earth to earth microwave signal attenuation began in the 1940s, using rain gauges along the link path in order to correlate uniform rainfall with signal attenuation, where the received power was measured at intervals of 30 s [9]. In recent years, most work focuses on measurements from existing commercial microwave links [9,10] and references therein.

The symbol error probability can be found as a function of the attenuation. Attenuation as a function of rain rate has been widely calculated and measured [7,8,9,10,11,12,13,14,15,16], with the standard statistical method being the ITU-R recommendations [17,18,19], where we focus on outdoor long-distance communication as relevant to satellite-ground links. The challenge is that the symbol error probability is a non-linear function in terms of signal-to-noise ratio and attenuation, which is correlated with the rain rate.

Therefore, the present statistical method, using the average attenuation, is less precise, especially for the high frequencies that will be used soon. It is inaccurate because it does not consider the dynamic change in rain rate, which may at one moment be 10 mm/h, then, a few seconds later, may drop to 5 mm/h, and so on. The dynamics imply that the rain rate changes during a single rain event. Where in a single event, the rain can begin weakly, strengthen, and then become weaker. We use the statistics of the dynamic changes to achieve greater imprecision in the calculation of error, the required transmitter power, and the level of interference.

In this work, we derive a closed mathematical formula for the probability of symbol error rate as a function of the rain rate probability density function (pdf). The unique aspect of our work is that we consider the dynamic nature of the rain rate during a short-term rain event. Since the symbol error rate is a non-linear function, the use of various accepted methods can be imprecise in some cases. In addition, our work differs from accepted statistical methods, such as the ITU-R recommendations, as follows:

ITU-Rec. 618-13 calculates long-term rain attenuation statistics from the point rainfall rate obtained through the power law [18]. In contrast, we are interested in the probability of symbol error for a particular rain event. The ITU recommendation uses the measured rainfall rate exceeding 0.01% of the time and then extrapolates attenuation at other time percentages. Therefore, the ITU method provides identical results for two sites whose rate values were exceeded 0.01% of the time, even if, at other percentages of time, they differed widely.

ITU-R618-13 uses specific attenuation for the calculation, as given by the ITU-R Rec. 838-3 [17]. This has been found to show good agreement with measured data at frequencies below 100 GHz. At higher frequencies, drop size distribution (DSD) becomes a significant factor, and physically based models using Mie scattering and the DSD models exhibit better fit to measured data [7,20], though experimental work at these frequencies is scarce [7]. DSD models include Marshall–Palmer [21], JOSS [22], and Weibull [23].

Our method employs the specific attenuation as given by the ITU-R Rec. 838-3 [17], however, in contrast, we use the probability distribution function for rain rate in order to develop a mathematical expression for symbol error rate. This term is a function of rain rate, mean and variance, specific attenuation, as well as the symbol SNR in the absence of rain.

An ITU recommendation also includes a dynamic approach, but it differs from ours. ITU-R Rec. P.1853-2 [19] provides methods for synthesizing time series of tropospheric impairments, including rain attenuation on earth–space paths as well as single terrestrial paths. There, attenuation is derived from the assumption that the long-term statistics for attenuation are the lognormal distribution and recommends using ITU-R P.618-13 [18] to obtain that. In contrast, our method relies on the assumption that the rain rate, rather than attenuation, is lognormally distributed, and the attenuation is obtained as a function of rate. ITU-R Rec. P.1853-2 involves a synthesis of a white Gaussian noise series and is a numerical procedure, while our method provides a closed mathematical term for symbol error rate.

ITU-R.Rec.837-7 [24] provides a method for predicting rainfall rate statistics with a one-minute integration time. This can be used to provide the rate and the mean in our method, since our method does not predict a specific rate, but rather the rate appears as a parameter in the final expression. Our work is based on a derivation of the probability distribution function [25], which also provided data for the mean and variance; thus, for purposes of this paper, those were used. We focus on outdoor long-distance communication as relevant to satellite-ground links. The challenge is that the symbol error probability (SEP) is a non-linear function in terms of signal to noise ratio and attenuation, which is correlated with the rain rate. Therefore, employment of the present statistical methods for rain rate and attenuation in order to calculate SEP is less precise, especially for the high frequencies that will be used in the near future.

In this work, we calculate the correct average symbol error probability. This could be a crucial improvement because, when using the standard method, the values of the symbol error rate (SER) under-estimate the performance in comparison to reality. With 6G technology and, especially, with extensive use of IoT, a precise calculation of error will be of overriding importance.

The paper is organized as follows. In the first section, some background is presented regarding rain and the statistics or rain rate. Next, the probability density function for rain is given. The introductory background concludes with details of rain attenuation. In the following section, we find a closed mathematical expression for the average symbol error as a function of the probability density function of rain. A numerical comparison of methods is presented, and the paper ends with a discussion of the results.

## 2. Rain Rate Statistics

Rain is generated when water vapor in the atmosphere condenses until it is heavy enough to fall to earth. There are two types of rain: the first is stratiform rain, with almost no vertical wind, covering an extended area and relatively homogenous on a horizontal level. The second is convective rain, with a vertical wind causing up and down draft and high rainfall rates. Rain will affect polarization of signals, as raindrops are not spherical and attenuate horizontally polarized waves more than vertically polarized waves; this must be considered when encoding signals using polarization. In predicting rainfall, one wishes to obtain the probability of rainfall and rain rate. Rainfall can be predicted by partial measurement, by statistical models, and by a combination of these [7,8,9,10,11,12,13,14,15,16]. Empirical models for the probability of rain were developed based on experimental data, valid only in specific climates or frequency ranges.

Numerical prediction methods have been developed, for example, MultiEXCELL [26] constructs rain cells and collects them to simulate the effects and rate of rainfall. There too, the rain cells’ probability of occurrence is analytically derived from the local statistics.

As detailed in the previous section, ITU-R.Rec.837-7 provides a method for prediction of rainfall rate with a one-minute integration time that is exceeded, for a given average annual probability of exceedance, for a given location. This is based on total monthly data on rainfall generated from two databases, over land and water, as well as monthly mean surface temperature data from ITU-R P.1510 [27]. The land database includes 50 years of data from the GPCC Climatology (V2015) database and, over water, 36 years of data from the European Centre of Medium-range Weather Forecast (ECMWF) Era Interim data. The recommendation provides digital maps of the annual rainfall rate data exceeded for 0.01% of an average year. Some studies display discrepancies between ITU-R results and measurements in certain climates, e.g., in the tropics [28] or South Korea [29]. In our paper, we assume the existence of the rain, so we deal with the symbol error rate in cases of rain.

It is generally accepted that rain attenuation follows a lognormal distribution [30], although, in tropical regions, the lognormal model is not adequate, and a gamma distribution—or a mixture of log normal and gamma—is a better fit [31]. Our model treats the rain rate as lognormal since the lognormal hypothesis has been shown to match well with measured data [18,19,24,25,26,27,28,29,30,31,32,33,34,35]. We utilize the lognormal distribution for the sake of definiteness; the model can easily be extended to the alternative distribution.

The probability density function for rain rate in this model follows a lognormal distribution:(1)$$f(r_{r})=\frac{1}{{\left(2\pi \right)}^{1/2}r_{r}\sigma_{r}}\exp\left[-\frac{1}{2}{\left(\frac{\ln r_{r}-m_{r}}{\sigma_{r}}\right)}^{2}\right]$$
where *r_r_* is the rain rate and *m_r_* and *σ*^2^*_r_* are the mean and variance of the transformed variable. The mean and variance are derived from the mean and variance of the initial variables of rain rate, *m_R_*, and *σ*^2^*_R_*, as follows:(2)$$m_{r}=\ln m_{R}-\frac{1}{2}\ln\left(1+\frac{\sigma_{R}^{2}}{m_{R}^{2}}\right)$$
(3)$$\sigma_{r}=\ln\left(1+\frac{\sigma_{R}^{2}}{m_{R}^{2}}\right).$$

The function that predicts the attenuation of the RF signal as a function of rain rate is described in the following section. Table 1 shows the large variation in values of parameters that are typical for certain regions [25].

Figure 1 shows plots of the pdf for several values of parameters in Table 1.

## 3. Attenuation of RF Signal by Rain

Theoretical models have been developed to predict impairment and simulate and evaluate performance of fade mitigation techniques. Signal attenuation due to rain increases in proportion to the rainfall intensity and is also related to signal frequency. Upon obtaining prediction of rain, the attenuation must be calculated. The best known and utilized models for prediction of rain attenuation are those of the International Telecommunication Union Radiocommunication sector, detailed in Section 1. The ITU formula for specific attenuation of rain is given by a power law relation
(4)$$\gamma_{r}\left(r_{r}\right)=kr_{r}{}^{\alpha }$$
where *γ_r_* is the specific attenuation (dB/km), *r* is the rain rate (mm/h), and *k* and *α* are frequency dependent coefficients given specifically in the recommendation [17], where they are calculated for linear and circular polarization dependent on path geometry (path elevation angle and polarization tilt angle). This model is defined for frequencies ranging from 1 to 1000 GHz. All these model assumptions include stationarity of rainfall.

At high frequencies, particle diameter is relatively large compared to wavelength. It was found [8] that rain attenuation increases in proportion to frequency for frequencies below approximately 50 GHz, but that there is a peak at around 100 GHz. At higher frequencies, the degree of attenuation remains constant or decreases. This work employed the ITU-R power law for calculation of attenuation. Other approaches based on Mie scattering found that the Weibull distribution [36] is suitable above 30 GHz, or later, that Best and P-S distributions are appropriate from 90 to 225 GHz while the ITU-R model and Mie scattering with Weibull are best at 313 and 355 GHz [8]. Table 2 shows parameters for the specific attenuation for various frequencies, as calculated with mathematics from the formula given in [17]. The computation is for circular polarization but can easily be utilized for specific linear polarization as well. Polarization tilt relative to horizontal and path elevation were taken as 45 degrees.

## 4. Mathematical Model of Symbol Error Rate

In many applications in the mm wave and terahertz range, QAM is the leading modulation format or is part of a more complicated modulation such as OFDM or MIMO, therefore, our synthesis considers a general model of the symbol error rate (SER) of M- QAM. There have been various works on the dependence of SER on rain attenuation; examples include [37,38,39]. We chose to use a textbook expression for symbol error rate independent of rain [40] in our model, and then to weight it with the probability for rain rate, which is widely considered to be lognormal, as noted above; though, as noted above, in some regions, another model may be a better fit. Our model can thus be adapted in a straightforward manner if another model for rain rate or SER were to be desired. The independent probability for symbol error is
(5)$$P_{M}\left(s\left(r_{r}\right),M\right)=2\left(1-\frac{1}{\sqrt{M}}\right)erfc\left(F\left(s\left(r_{r}\right),M\right)\right)\times \;\left[1-\frac{1}{2}\left(1-\frac{1}{\sqrt{M}}\right)erfc\left(F\left(s\left(r_{r}\right),M\right)\right)\right]$$
where *erfc* is the complementary error function, defined as 1 minus the error function, or as *erfc*(*z*) = $2/\sqrt{\pi }\int_{z}^{\infty }exp\left[-t\right]dt$, and *F* and *s* are defined as follows:(6)$$F\left(s\left(r_{r}\right),M\right)=\sqrt{\frac{3}{2\left(M-1\right)}ns\left(r_{r}\right)}$$
(7)s(rr)=s010−γ(rr)z010

Here, *M* = 2*^n^*, *s*_0_ is the average SNR per bit without rain, γ is the specific attenuation of rain from (4), and *z*_0_ the distance by which the signal passes through the rain (km). This can be approximated by [30]
(8)$$P_{M}\left(s\left(r_{r}\right),M\right)\leq 2erfc\left(F\left(s\left(r_{r}\right),M\right)\right)$$
thus providing an upper bound for the error. However, we wish to take into account the probability for rain. Using the expression for lognormal distribution as found in (1) and the term for symbol error probability as in (5)–(8), the average symbol error probability of M-ary QAM in this case is given by multiplying the symbol error probability *f* as a function of rain rate by the probability distribution function for rain rate, and integrating overall rain rates:(9)$$\bar{P_{M}}=\int_{0}^{\infty }P_{M}\left(s\left(r_{r}\right),M\right)f_{r}\left(r_{r}\right)dr_{r}.$$

We are looking for a closed form expression for the probability of symbol error. The ITU-R attenuation is in dB/km, and we convert it in the following calculation. The minus sign is used to indicate that this is a loss. Collecting constants and plugging in (2) and (5)–(8), this is written explicitly as
(10)PM¯=∫0∞Arrerfc(Bs010−krrα10)exp[−12(lnrr−mrσr)2]drr
where *A* is the first term of the lognormal distribution, and *B* collects constants from (6).
(11)$$A=\frac{2}{\sqrt{2\pi }\sigma_{r}},B=\sqrt{\frac{3n}{2\left(M-1\right)}},$$

Writing out the complementary error function as an integral and collecting exponential terms:(12)$$\bar{P_{M}}=\frac{2A}{\sqrt{\pi }}\int_{0}^{\infty }\left(\frac{1}{r_{r}}\int_{C_{2}\sqrt{s10^{kr_{r}^{\alpha }}}}^{\infty }\exp\left[-\frac{1}{2}{\left(\frac{\ln r_{r}-m_{r}}{\sigma_{r}}\right)}^{2}-t^{2}\right]dtdr_{r}\right).$$

In order to achieve a more tractable expression, we perform the following substitution:(13)y=t−Bγb10−arrb10

Here, *dy* = *dt*. We exchange the integration order, so we first integrate over *dy* and, only afterwards, over *dr*:(14)PM¯=2Aπ∫0∞(∫0∞1rrexp[−12(lnrr−mrσr)2−(y+Bsb10−krrα10)2]drrdy)

The term with the square root can temporarily be collected into a constant *C* that is not a function of *y*, enabling a simpler integration over *y*. Thus, the equation can be written as
(15)$$\bar{P_{M}}=\frac{2A}{\sqrt{\pi }}\int_{0}^{\infty }\left(\int_{0}^{\infty }\frac{1}{r_{r}}\exp\left[-\frac{1}{2}{\left(\frac{\ln r_{r}-m_{r}}{\sigma_{r}}\right)}^{2}-{\left(y+C\right)}^{2}\right]dr_{r}dy\right).$$

Integration over *y* gives
(16)$$\bar{P_{M}}=2A\int_{0}^{\infty }\frac{1}{r_{r}}erfc\left[C\right]\exp\left[-\frac{1}{2}{\left(\frac{\ln r_{r}-m_{r}}{\sigma_{r}}\right)}^{2}\right]dr_{r}.$$

Further, plugging constants back in, the average probability becomes
(17)$$\bar{P_{M}}=\frac{4}{\sqrt{2\pi }\sigma_{r}}\int_{0}^{\infty }\frac{erfc\left[C\right]}{r_{r}}\exp\left[-\frac{1}{2}{\left(\frac{\ln r_{r}-m_{r}}{\sigma_{r}}\right)}^{2}\right]dr_{r}$$
(18)$$C=\sqrt{\frac{3s}{2\left(M-1\right)}}\exp\left[-\left(\frac{k\ln[10]}{10}\right)r_{r}^{\alpha }\right]$$

In order to write the constant *C*, we have used
(19)s10−krrα10=exp[ln(10−krrα10)]=exp[−krrα10ln(10)].

Therefore, the complete term for the probability of symbol error is
(20)$$\bar{P_{M}}=\frac{4}{\sqrt{2\pi }\sigma_{r}}\int_{0}^{\infty }\begin{array}{l} \frac{erfc\left[\sqrt{\frac{3s}{2\left(M-1\right)}}\exp\left[-\left(\frac{k\ln[10]}{10}\right)r_{r}^{\alpha }\right]\right]}{r_{r}}\times \\ \times \exp\left[-\frac{1}{2}{\left(\frac{\ln r_{r}-m_{r}}{\sigma_{r}}\right)}^{2}\right]dr_{r} \end{array}$$

This can be solved numerically, but there is also a closed form in terms of Hermite polynomials. We substitute
(21)$$\begin{array}{l} \frac{\ln(r_{r})-m_{r}}{2\sigma_{r}}=t\\ dr=2r\sigma_{r}dt,r=\exp\left[2t\sigma_{r}+m\right] \end{array}$$
and insert this into (20). We obtain
(22)$$\bar{P_{M}}=\frac{4}{\sqrt{2\pi }}\int_{-\infty }^{\infty }\begin{array}{l} \frac{erfc\left[\sqrt{\frac{3s}{2\left(M-1\right)}}\exp\left[-\left(\frac{k\ln[10]}{10}\right)\exp{\left[2t\sigma_{r}+m\right]}_{}^{\alpha }\right]\right]}{\exp\left[2t\sigma_{r}+m\right]}\times \\ \times \exp\left[-t^{2}\right]dt \end{array}.$$

This takes the general form of
(23)$$\int_{-\infty }^{\infty }\exp\left[-t^{2}\right]f(t)dt$$
which can be expressed as
(24)$$\int_{-\infty }^{\infty }\exp\left[-t^{2}\right]f(t)dt=\sum_{i=1}^{n}w_{i}f\left(t_{i}\right)+R_{n}$$
where *t_i_* is the *i*th zero of the Hermite polynomial *H_n_*(*t*), and *w* is the weight:(25)$$\begin{array}{l} w_{i}=\frac{2^{n-1}n!\sqrt{\pi }}{n^{2}{\left(H_{n-1}\left[t_{i}\right]\right)}^{2}}\\ R_{n}=\frac{n!\sqrt{\pi }}{2^{n}\left(2n\right)!}f^{\left(2n\right)}\left[\xi \right],\begin{array}{cccc} & & & \left(-\infty <\xi <\infty \right) \end{array} \end{array}$$

Both the weights and the zeros, *w_i_*, and *t_i_* may be obtained from tables in [41].

## 5. Rain Rate Statistics

We performed a numerical calculation of the probability for signal error weighted by the pdf for rain (Equation (20)), and we compared this to the probability without considering the pdf for rain. The probability as developed in Equation (20) was numerically calculated for representative values of 64 QAM (*n* = 6 in the equations.) For the constants, we chose values of *k* and *α* from the ITU recommendation (ITU 838-3). We present here the results for 80 GHz, since our interest is in high frequencies. However, the same calculation was performed for additional frequencies giving similar results. Distance *z*_0_ was taken as 1 km, as representing a variety of applications, and *γ*_0_ taken as 500. We performed the simulation using alternate values and frequencies and found that it did not bring forth a significant difference in the results. In order to complete the simulation using concrete values for rain rate (*m_r_*, from Equation (20)), it was necessary for numerical purposes to create an array of values. Therefore, a vector was created for *m_r_* ranging from 1 to 300 mm/h, giving a great range of possible values for rain rate. Accordingly, a vector of equal length for standard deviation was obtained as follows: actual measured values for m and σ were taken from [26], and the average ratio of *m*/*σ* was calculated for the wide variety of values found there, with synthetic extrapolations to further explore the validity of the results. (See also [42,43,44,45].) It was found that, on average, *σ* = *m*/0.5. This was used to create a vector for σ matching the rate vector. These were then given logarithmic transformation as in (2) and (3), above.

The result of the simulation is shown in Figure 2, which presents a logarithmic plot of the probability for symbol error with and without considering the rain rate. In the figure, the log of the average probability (red solid line) is shown and compared to the probability as calculated in our model, where the pdf for rain is taken into account. The interesting part of the graph is on the left, for low rain rate, as the operating region for effective signal transmission is for a rain rate of less than 15 mm/h. In this regime, we see that without taking the probability of rain into account, the probability of symbol error is negligible, as shown in above. However, if we weight this with the pdf for rain, as is done in Equation (20), the probability for symbol error is considerably greater. Therefore, it appears vital to take into account the probability for rain rate in considering symbol error. This trend is reversed for rain rates of 30 mm/h and greater, but for such high rain rates, in any case, the error probability is too high for useful transmission. Simulations for a range of other frequencies appear in Figure 3 and show similar disparities in the relevant rain rates of under 40 mm/h.

Table 3 shows the SNR required for various error probabilities in both cases, for a rain rate of 10 mm/h for the given set of parameters. The main conclusion from the table is that design of the link without using the correct method could result in a loss of 3 dB or more in some cases.

## 6. Conclusions

We have developed a method for calculating the probability for symbol error rate that considers variations in rainfall rate. This is in contrast to accepted methods which employ average values over time. We also gave a closed-form mathematical expression for the probability of error. Numerical calculation clearly shows that when the density distribution function of rain rate is considered, the probability of symbol error rate changes significantly. This model will enable the design of a system that will ensure the desired bit error rate, for example, by adapting the power transmitter, modulation constellation, and/or coding strength.

Our aim in this work has been to include the rain rate probability, which is a major attenuating factor in signal transmission, in the symbol error rate calculation. There are other attenuating factors, including clouds, atmospheric gases, dust, and sandstorms. For the higher frequencies under discussion, the relation of wavelength to particle size makes these too increasingly significant in contributing to signal error. Attenuation by these has been calculated in various ways (see for instance [46]). The wave scattering by small raindrops could cause higher attenuation. If a modified specific attenuation model is available, it could be used in Equation (9) instead of the present one. Future work on higher frequencies should include models of specific attenuation that take DSD into account. In addition, future work could also extend our calculation to consider the probability of other atmospheric impairments.

Future work should consider troposphere scintillation. When the wind blows, the mainly horizontal layers of the equal refractive index in the troposphere tend to become mixed due to turbulence. The troposphere being mixed leads to rapid refractive index variations over small distances or scale sizes and over short time intervals. Waves traveling through these rapid variations in the index vary in amplitude and phase. This situation creates dry tropospheric scintillation. Another source of tropospheric scintillation is rain, which tends to occur at slower rates than the dry effects. Scintillation is not an absorbent effect in that the signal’s mean level is unchanged. The effect is strongly frequency-dependent in that shorter wavelengths will encounter more severe variations resulting from a given scale size [47].

With 6G on the horizon and communication that will be used not only among people but with devices that may have life-or-death importance, the need for an exact calculation of symbol error rate is imperative. In addition, 6G will use very high frequencies, which make it particularly susceptible to attenuation and distortion by rain. Therefore, accurate calculation of symbol error will be necessary with the developing communication technology.

## Figures and Tables

**Figure 1 sensors-22-06269-f001:**
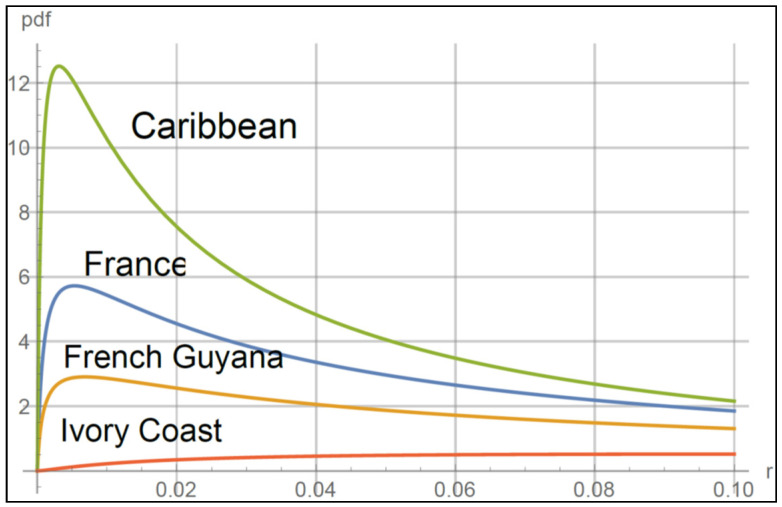
Lognormal distribution for values of parameters in Table 1.

**Figure 2 sensors-22-06269-f002:**
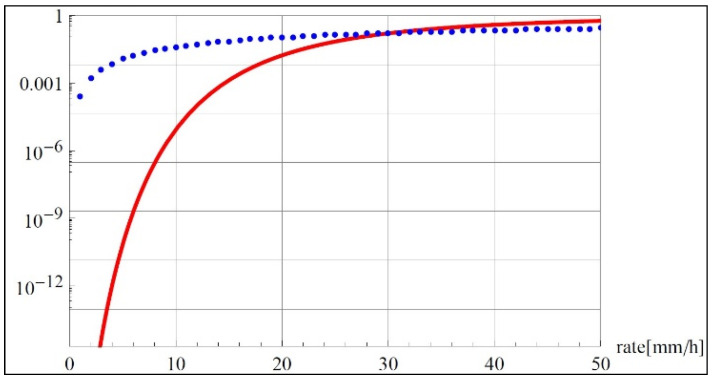
Logarithmic plot comparing probability for symbol error. The red solid line is the log of average probability as in (6), and the blue dotted line is the probability considering the pdf for rain, (20). For low rain rate, which would be the practical range for signal transmission, the probability for error is higher if the rain rate is considered. The calculation was performed for *M* = 64, *f* = 80 GHz.

**Figure 3 sensors-22-06269-f003:**
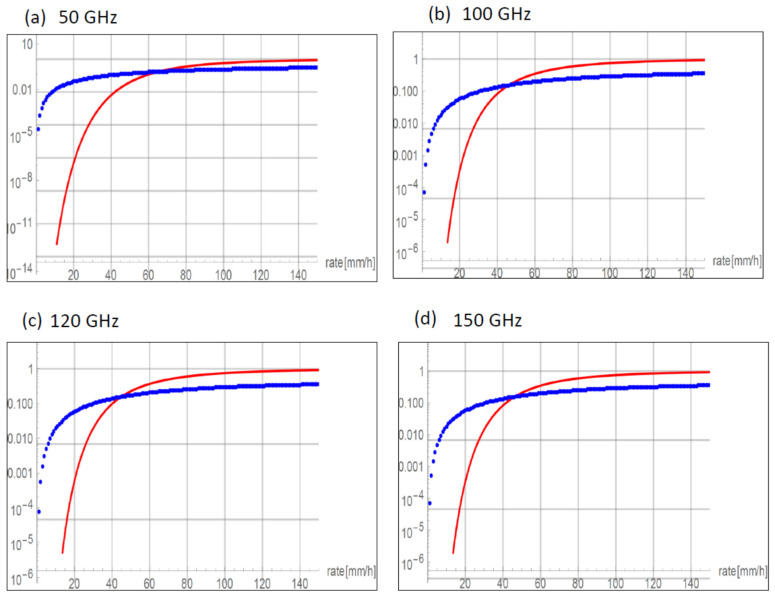
Logarithmic plot comparing probability for symbol error for a range of frequencies. The red solid line is the log of average probability as in (6), and the blue dotted line is the probability considering the pdf for rain, (20). For low rain rate, which would be the practical range for signal transmission, the probability for error is higher if the rain rate is considered. The calculation was performed for *M* = 64 and for (**a**) 50 GHz, (**b**) 100 GHz, (**c**) 120 GHz, and (**d**) 150 GHz. For higher frequencies, atmospheric disturbance will be more significant, as discussed in the text.

**Table 1 sensors-22-06269-t001:** Typical parameters for various regions.

Region	*σ* ^2^ * _R_ *	*σ* ^2^ * _r_ *	*m_R_* (mm/h)	*m_r_* (mm/h)
Southwest France	2.38	1.95	0.63	−1.43
Ivory Coast	108	1.74	4.81	0.7
Caribbean	23	2.12	1.77	−0.49
French Guyana (equatorial coastal)	0.42	1.85	0.24	−2.35

**Table 2 sensors-22-06269-t002:** ITU parameters for attenuation constants.

	**mm Band**	
**Freq (GHz)**	** *k* **	** *α* **
30	0.2347	0.931125
80	1.1686	0.706807
100	1.36755	0.678999
	**Terahertz Band**	
**Freq (THz)**	** *k* **	** *α* **
0.2	1.64105	0.636246
0.3	1.6286	0.6279
0.4	1.584	0.6259
0.7	1.4638	0.629948
0.8	1.4328	0.63245
1	1.38085	0.637598

**Table 3 sensors-22-06269-t003:** SNR for various probabilities for a rain rate of 10 mm/h for 80 GHz.

Error Probability	SNR Required for Classical Case (dB)	SNR Required (dB)	Difference (dB)
10^−4^	26	23	3
10^−5^	26.5	24.8	1.7
10^−6^	27.8	26	1.8
10^−7^	28.4	27.4	1

## Data Availability

Not applicable.

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
