# Peer review of "Communication Systems Performance at mm and THz as a Function of a Rain Rate Probability Density Function Model"

_sensors, 2022, doi:10.3390/s22166269_

Round 1

Reviewer 1 Report

Although mathematical formulation must be correct as presented, some simplified calculations would be helpful for the readers, who have less deep mathematical background. Additionally analytical results should be extended with supporting simulations or measurements.

One small typo: Millimeter should be abbreviated as mm (not MM).

Author Response

We thank the reviewer for the comments and suggestions.

We have added explanations and clarifications in the text between the equations (5) through (25.) We hope this is clearer now.

MM has been changed to mm where it appears, in Table 2.  Additional simulations have been performed and are shown in the new Figure 3.

Reviewer 2 Report

Reviewer’s Recommendation

This manuscript needs major revisions. I like this work but further work should be done.

Summary

The manuscript deals with communications in mm and THz under rain conditions.

General comments

The manuscript has some interesting facts to show but at the beginning (introduction) more works should be correlated under this proposed technique (please see next section) thus a more analytical expression is needed relevant to mathematics. Also, more measurements are needed to validate your work. Please do so.

Suggested Improvements

In addition to the previously mentioned, please apply the following suggestions/addons, and import the answers inside the revised manuscript:

  1. The abstract should contain also the universal factor of the proposed work. E.g. How this can be implemented in future and existing systems (in brief).
  2. The introduction lacks of showing some historical elements based on research works and how they would affect them. Please see (1) Earth-to-Earth Microwave Rain Attenuation Measurements: A Survey On the Recent Literature and (2) Rain attenuation at millimeter wave and low-THz frequencies, (3) Atmospheric attenuation in wireless communication systems at millimeter and THz frequencies [wireless corner], (4) A rain estimation model based on microwave signal attenuation measurements in the city of Ioannina, Greece and (5) Secrecy performance of terahertz wireless links in rain and snow
  3. "Our model treats the rain rate as lognormal. our model we utilize the lognormal distribution". Please show the mathematical analysis procedure that leads you to your proposed method.
  4. Equations (5) to (25) need somehow better explanations. Please elaborate. E.g. Give in-between equations some adequate explanations.
  5. In figure 2 you mention that "The calculation was done for M=64, f=80GHz". Please elaborate on why you have chosen that frequency. At these levels what about Tropospheric Scintillation? Please explain. E.g. please see Tropospheric scintillation with concurrent rain attenuation at 50 GHz in Madrid.
  6. Conclusion somehow comes relevantly quickly. I propose that before concluding your work to further check with more measurements in order to show us a true "stress-test" of your work that will indeed help to diffuse your research project.

Manuscript Rating:

1.    This manuscript needs major revisions.

2.    A future implementation using automatic routines and procedures should be accomplished.

Reviewer 3 Report

I consider that there could be more than one set of data compared to the results obtained for the formulas presented. Also, the data obtained for more  than one frequency should be specifically presented in order to increase the credibility of the study.

Another thing that could be improved is that there are some abbreviations that are not explained (some of them are explained only when they are used the second time, some are not explained at all ). 

Formulas must be also rewritten and their numbers should be aligned.

Round 2

Reviewer 2 Report

The manuscript was adequately revised. Please take into consideration correcting some grammatical and expressional mistakes in advance.

Please recheck the manuscript for some minor mistakes at the mathematical and presentation levels. No further reviewing is needed on my part.

Good luck with your work.